# Auto-amplification and spatial propagation of neutrophil extracellular traps
Pan Deng[1,2], Alec Xu[1,2], Peter M. Grin [2,3], Kerryn Matthews[1,2], Simon P. Duffy[1,2,4] & Hongshen Ma [1,2,5,6] ✉

The release of cellular DNA as neutrophil extracellular traps (NETs) plays a pivotal role in the immune response to pathogens by physically entrapping and killing microbes. NET release occurs at a greater frequency within neutrophil clusters and swarms, indicating a potential for collective behavior. However, little is known about how dense clustering of cells influences the frequency of NET release. Using an image-based assay for NETosis in nanowells, we show that the frequency of NETosis increases with cell density. We then co-incubate NETotic neutrophils with naïve neutrophils and find that NETotic neutrophils can induce secondary NETosis in naïve neutrophils in a cell density-dependent manner. Further mechanistic studies show that secondary NETosis is caused by a combination of DNA and protein factors. Finally, we immobilize NETotic neutrophils in a plaque, and then place the plaque near naïve neutrophils to characterize the spatial propagation of secondary NETosis. We find that secondary NETosis from naïve neutrophils increases over time, but remains spatially restricted to the periphery of the plaque. Together, we show that NETosis is an auto-amplified process, but that the spatial propagation of NET release is strictly regulated.

Neutrophils are the most abundant type of circulating immune cells in healthy humans. These cells patrol the body to control infections and remove dead cells using strategies including phagocytosis, degranulation, and release of neutrophil extracellular traps (NETs)[1,2]. NETs are web-like structures of DNA decorated with enzymes and citrullinated histones, which have the ability to trap and kill invading pathogens[3]. The release of NETs constitutes the last step in a deliberate cell death process called NETosis, which can be initiated through multiple pathways including Toll-like receptors (TLRs) that detect pathogen-associated molecular patterns (PAMPs) such as lipopolysaccharide (LPS) as well as damage associated molecular patterns (DAMPs) resulting from cell death, Fc-receptors that detect antibody-bound cells and receptors for inflammatory cytokines (e.g. LTB4 and interleukin-8)[4]. Intracellular signaling from these pathways ultimately converges on the activation of myeloperoxidase (MPO), neutrophil elastase (NE), and protein-arginine deiminase type 4 (PAD4)[5]. PAD4 contributes to chromatin decondensation, while the cell undergoes lysis to release both DNA and citrullinated histones[6,7].

In addition to direct stimulation by microbial components and signaling factors, NETosis is also potentiated when neutrophils act as a collective within functional cell clusters. An example of a functional cell cluster is a neutrophil swarm, where neutrophils engage in collective behavior to efficiently eliminate pathogens[8]. These swarms release NETs[9–11], which enhances the persistence of neutrophil swarms[12]. While the collective behavior of neutrophils in swarms has been previously described, relatively little is known about whether neutrophil clustering and collective behavior influence the frequency or propagation of NETosis. Importantly, it is not known whether NETosis is purely a direct response to PAMPs and DAMPs, or if NETosis can be propagated from stimulated neutrophils to naïve neutrophils via cell-cell signaling. NET-induced NETosis has been reported previously, but was found to be a consequence of mechanical disruption of NETs due to injury[13]. If NETosis can be propagated to naïve neutrophils in swarms and cell clusters, does this response enable the enlargement of NETs from the original nidus of infection to prevent pathogen escape?

Here, we investigate the interactions of naive, unstimulated neutrophils with nearby neutrophils undergoing NETosis. First, we used nanowell

[1]Department of Mechanical Engineering, University of British Columbia, 2054-6250 Applied Science Lane, Vancouver, BC, Canada. [2]Centre for Blood Research, University of British Columbia, 2350 Health Sciences Mall, Vancouver, BC, Canada. [3]Department of Biochemistry & Molecular Biology, University of British Columbia, 2350 Health Sciences Mall, Vancouver, BC, Canada. [4]British Columbia Institute of Technology, 3700 Willingdon Avenue, Vancouver, BC, Canada. [5]School of Biomedical Engineering, University of British Columbia, 2222 Health Sciences Mall, Vancouver, BC, Canada. [6]Vancouver Prostate Centre, Vancouver General Hospital, 2660 Oak Street, Vancouver, BC, Canada. ✉e-mail: hongma@mech.ubc.ca

confinement to investigate whether NETosis depends on neutrophil density. Second, we used a similar approach to investigate whether NETosis can propagate from stimulated cells to proximal naïve cells, and to determine which components are important in such propagation. Finally, we developed an assay to investigate whether NETosis can propagate from a central plaque to peripheral naïve neutrophils in neutrophil clusters.

## Results

### NETosis depends on cell density

To determine whether NETosis depends on cell density, we developed a nanowell NETosis assay, where we can track NETosis for a small number of cells in each nanowell. We fabricated nanowells with dimensions of 80 μm × 80 μm × 35 μm ($l × w × h$) inside standard 384-well imaging microwell plates using laser micropatterning of poly(ethylene glycol) diacrylate (PEGDA) hydrogel (Fig. 1a)[14]. Each microwell contained ~2000 nanowells. NETosis was induced using ionomycin treatment of HL-60 promyelocytic cells, which were differentiated into a neutrophil-like state (dHL-60)[15]. Cells were imaged at multiple time points and NETosis was observed based on Hoechst staining of DNA (Fig. 1b). Specifically, NETosis cells appear to have diminished intensity, as well as the appearance of the

DNA stain to extend beyond the cell boundary. To confirm that these properties were indeed driven by NETosis, we used antibodies to stain for cellular release of myeloperoxidase and neutrophil elastase, which are characteristic of NETosis[5] (Fig. 1c).

To investigate the relationship between cell density and NETosis, ionomycin stimulated cells were randomly distributed in nanowells, at a density of 0–30 cells per nanowell. The frequency of NETosis in each well was quantified as a percentage of intact cells before and after the incubation period. To detect NETosis cells, we first captured initial images of stimulated cells to obtain a threshold for detecting intact cells. After the incubation period, we counted the number of intact cells remaining within three independent nanowell-in-microwells in order to determine the number of cells that have undergone NETosis in each well. In the absence of ionomycin, we observed that a baseline frequency of NETosis in dHL-60 cells did not increase with cell density. With ionomycin stimulation, the frequency of NETosis ranged from 25–69% and exhibited a linear correlation between cell density within a nanowell and the percentage of cells undergoing NETosis ($R^2 > 0.76$) (Fig. 1d). Similar correlations were observed in primary human neutrophils that were treated with either ionomycin ($R^2 > 0.76$) or LPS ($R^2 > 0.69$) (Fig. 1e). These results show that increased cell density is

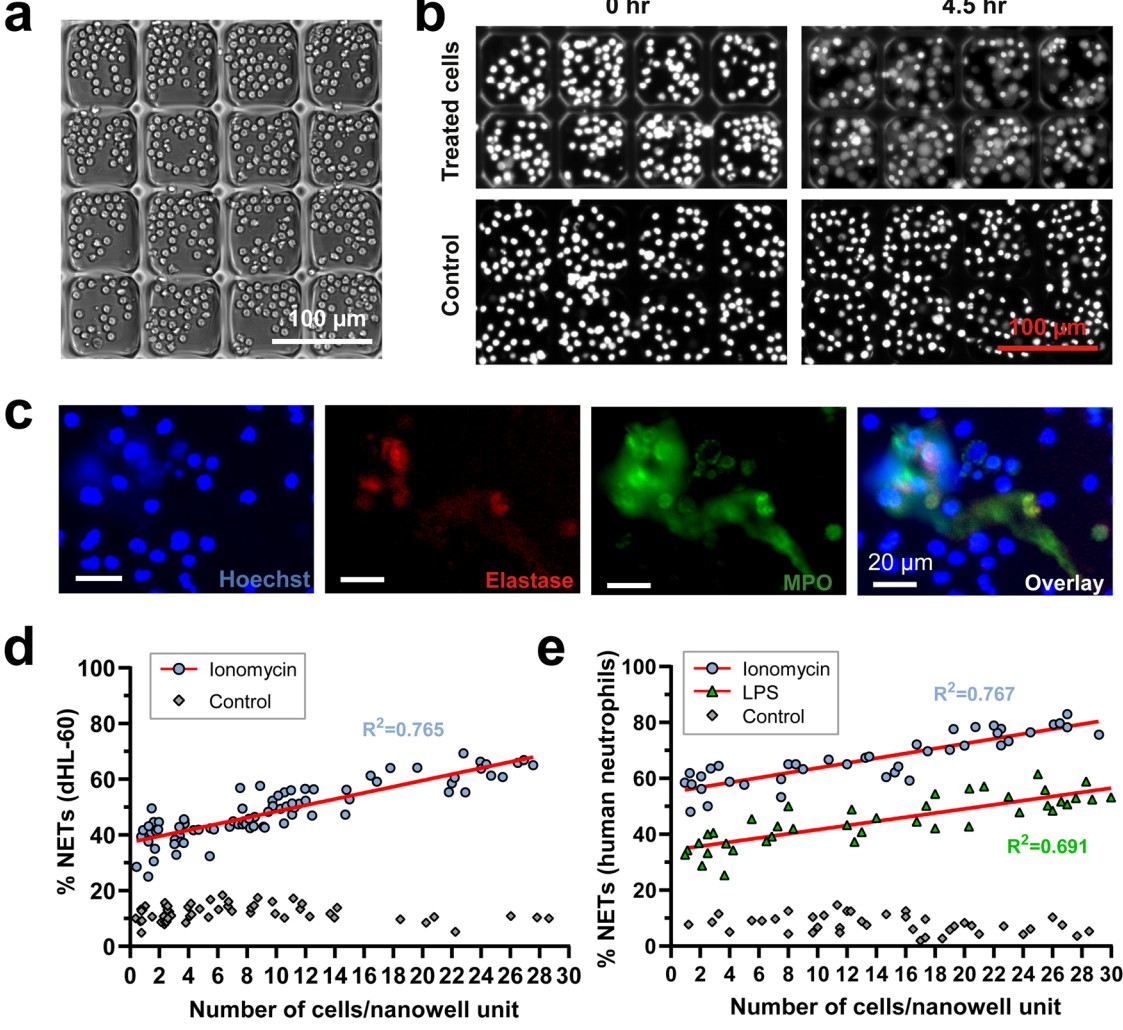

**Fig. 1 | NETosis depends on cell density. a** A representative image of naïve dHL-60 cells incubated in nanowells formed by laser micropatterning of poly(ethylene glycol) diacrylate hydrogel within one microwell of a 384-well plate. Scale bar: 100 μm. **b** dHL-60 cells stained with Hoechst were treated with either ionomycin (5 μM) or culture medium (control) at baseline (0 h) and after 4.5 h. Scale bar: 100 μm. **c** Representative microscopy images showing NET induction in dHL-60

cells treated with ionomycin (1 μM) for 2 h. Scale bar: 20 μm. **d** Percentage of dHL-60 cells producing NETs after ionomycin stimulation (6 μM, 4.5 h) relative to the averaged density of total cells incubated in nanowells. **e** Percentage of human neutrophils producing NETs after ionomycin (5 μM, 4.5 h) or LPS stimulation (20 μg/mL, 4.5 h) relative to the averaged density of total cells incubated in nanowells. All P values < 0.001.

associated with increased NETosis frequency, which suggests the potential for cell-cell signaling between nearby cells, through either direct cell-cell contact or cell-NET contact, or through secreted factors, to enhance NETosis. If such signaling exists, then it may be possible for NETotic neutrophils to trigger secondary NETosis in naïve neutrophils.

## Secondary NETosis

To investigate whether NETotic cells can induce secondary NETosis in nearby naïve cells, we co-incubated ionomycin-stimulated dHL-60 cells with naïve dHL-60 cells in nanowells (Fig. 2a). We used ionomycin to stimulate primary NETosis because ionomycin treatment induced NETosis in a greater fraction of cells when compared with lipopoly-saccharide (LPS) or phorbol 12-myristate 13-acetate (PMA) (Supplementary Fig. 1). The ionomycin was then removed by washing twice. To confirm the effective removal of ionomycin, the supernatant from the second wash was tested on naïve dHL-60 cells, which did not induce NETosis (Supplementary Fig. 2). After washing, the stimulated dHL-60 cells were then added to naïve cells in nanowells and imaged at 0 and 4.5 h to assess secondary NETosis (Fig. 2a). We found that stimulated dHL-60 cells were able to induce secondary NETosis in naïve dHL-60 cells, and the fraction of cells undergoing secondary NETosis increased with the number of ionomycin treated cells (Fig. 2b). We then repeated this experiment using primary neutrophils from healthy human donors. Stimulated primary neutrophils also induced secondary NETosis in naïve primary neutrophils, and the fraction of secondary NETosis cells was similarly dependent on the number of stimulated cells in each

nanowell (Fig. 2c). These results confirm that NETotic neutrophils can induce secondary NETosis in proximal naïve neutrophils.

To investigate the mechanism for secondary NETosis, we first studied the neutrophilic response to cell-free DNA via Toll-like receptor 9 (TLR9)[16–18] signaling. We found that using the oligonucleotide ODN-A151 to block the TLR9 receptor, in the presence of stimulated cells, reduced secondary NETosis relative to the control oligonucleotide (Fig. 2d). This result suggests that secondary NETosis is at least partially mediated by TLR9 and sensitive to extracellular DNA. We then tested the effect of adding DNase I to stimulated neutrophils. DNA digestion resulted in further reduction of secondary NETosis, which also confirmed that secondary NETosis is mediated by extracellular DNA. These results, however, do not indicate whether cell-free NETs are sufficient to induce NETosis, independent of other cell signals. To evaluate this further, we treated naïve cells with cell-free NETs, which were isolated from a suspension of ionomycin-treated cells. Interestingly, cell-free NETs could not initiate secondary NETosis in naïve cells (Fig. 2d), which suggests that an additional cell-associated factor is required for secondary NETosis. To further narrow the range of molecular mechanisms responsible for secondary NETosis, we treated stimulated cells using Proteinase K to digest protein components associated with these cells. After digestion, the stimulated cells were heated to 55 °C to inactivate the Proteinase K, and then co-incubated with naïve cells as before. Proteinase K treatment also eliminated the secondary NETosis in naïve neutrophils (Fig. 2d), which indicates that secondary NETosis also requires either NET-associated proteins or NETotic cells. Together, these results show that secondary NETosis in naïve neutrophils is

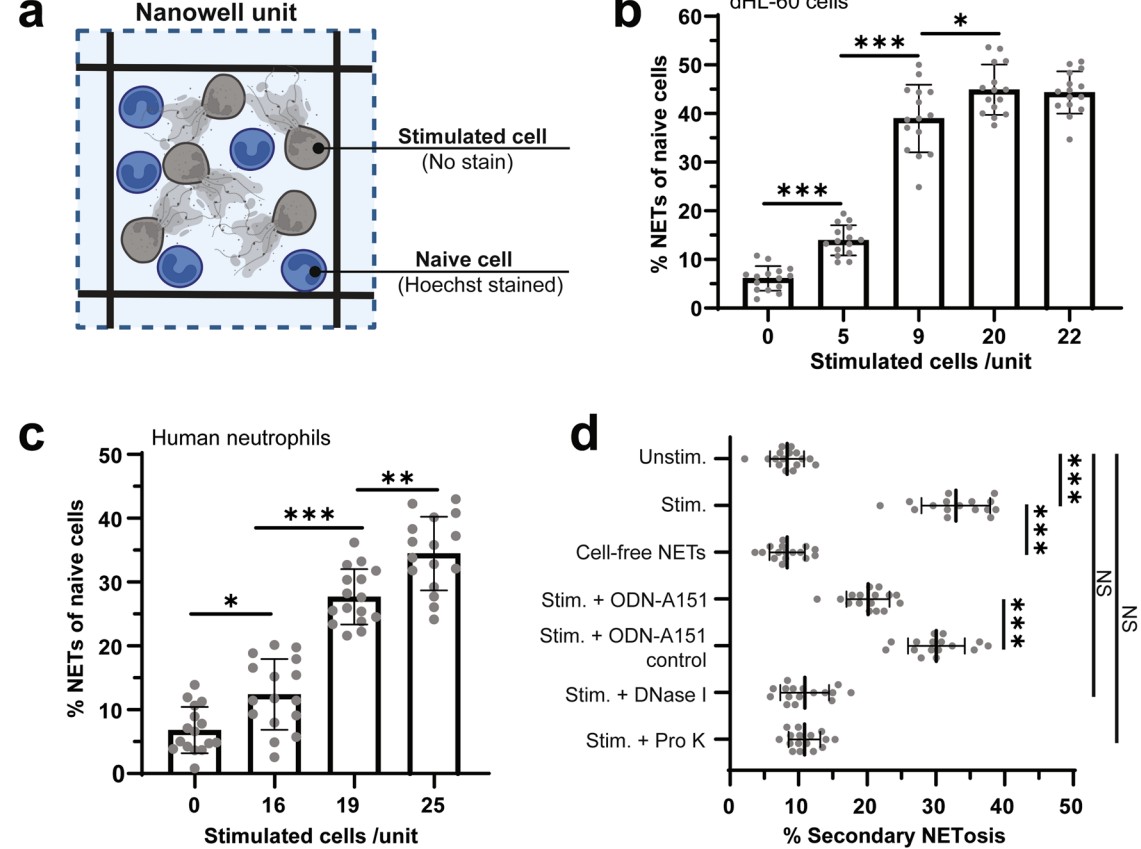

**Fig. 2 | Secondary NETosis in nanowells. a** Schematic of the secondary NETosis assay in nanowells where naïve cells were incubated with cells pre-stimulated with ionomycin (6 μM) in clean media. The secondary NETosis was quantified by the measurement of NETs release in naïve cells: The NETs release was evaluated by Hoechst staining of naïve cells and measured as the percentage of cells undergoing NETosis. NETs release by (**b**) dHL-60 cells and (**c**) primary neutrophils after co-incubating with ionomycin-stimulated cells at varying cell densities. **d** Secondary NETosis in primary neutrophils after different treatments on stimulated cells: stimulated cells (no treatment), cell-free NETs isolation, ODA-A151 treatment, DNase I treatment, Proteinase K treatment. All statistical analyses were carried out using one-way ANOVA with Tukey's post-hoc test *$p < 0.05$; **$p < 0.01$; ***$p < 0.001$. All the values shown were averaged from three individual experiments. Error bars represent standard deviation.

induced by proximal NETotic neutrophils through a combination of DNA and protein factors.

## Auto-amplification and Spatial propagation of NETosis

Our findings suggest that NETotic neutrophils may be able to auto-amplify NETosis in a cell cluster, thereby triggering secondary NETosis in proximal neutrophils to propagate NETosis throughout the cluster. Secondary NETosis initiated by contact between NETotic neutrophils and naïve neutrophils is a potential auto-amplification mechanism that allows NETs to propagate from its triggering point at the nidus of infection towards distal neutrophils arriving after migration. We developed an in vitro assay to investigate this possibility. We first generated a NET plaque by immobilizing ionomycin-treated dHL-60 cells on chromatography paper (Fig. 3a, Supplementary Fig. 3). The dHL-60 cells were concentrated in an ~2 mm diameter disk and residual ionomycin was removed by washing. As before, the washed supernatant did not trigger NETosis. Next, the NET plaque along with the chromatography paper was gently overlaid upside down on a layer of naïve dHL-60 cells in a flat-bottom glass microwell plate. To track the NET plaque and naïve cells over time, the NET plaque was stained with Hoechst (blue) while the naïve cells (untreated cells) were stained with CellTracker Red. To visualize NETosis, SYTOX Green, a membrane-impermeable DNA dye was included in the media. We imaged the sample every 40 min after the NET plaque came in contact with naïve cells. Initially, the naïve cells showed minimal NETs, both in regions near the NET plaque (c2) and farther away from NET plaque (c1) (Fig. 3b–e). After 180 min, a significant amount of NETs were observed, with greater NETosis nearer the NET plaque (d2) compared to farther away from the NET plaque (d1) (Fig. 3d). In control experiments, where the NET plaque was replaced with unstimulated cells, few NETs were observed over the duration of the experiment (Fig. 3e). These results confirm that NETosis from stimulated cells in a plaque can trigger NETosis in nearby naïve cells, which forms a positive feedback loop that auto-amplifies the spatial propagation of NETosis from the original stimulated group of neutrophils.

## Quantification of NETosis propagation

We developed an image analysis approach to quantify the spatial propagation of naïve cell NETosis originating from a NET plaque. We prepared for this image analysis by first excluding the stimulated cells in the central NET plaque, which is identified by pixels stained by Hoechst (Fig. 4a). Next, we removed dead cells that retain their DNA using an intensity threshold for

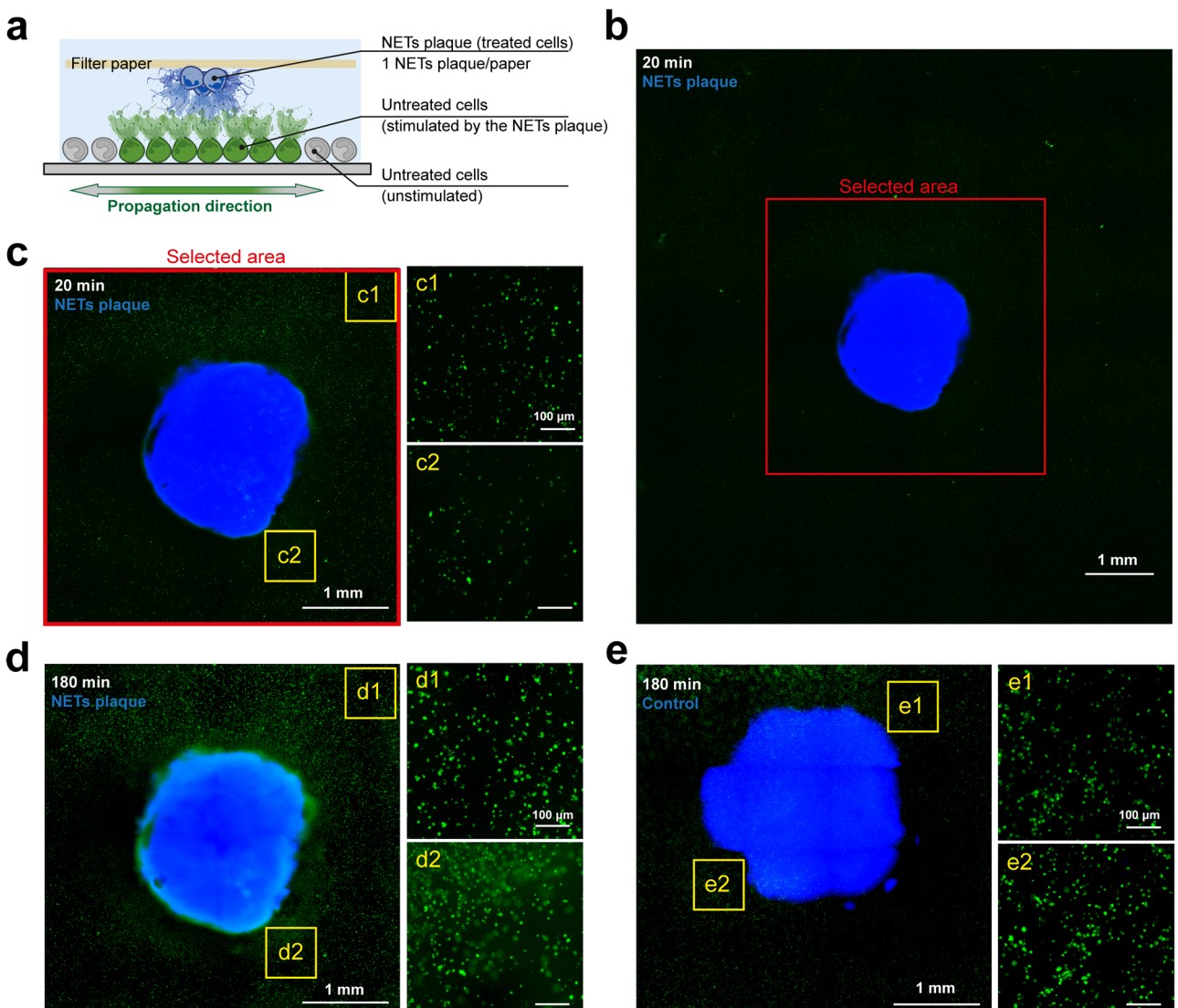

**Fig. 3 | NETosis propagation starting from a plaque of stimulated cells.**
**a** Schematic of the NETosis propagation assay. **b**–**e** Representative images of NETosis propagation at 20 min (**b**, **c**) and 180 min (**d**) after exposure to stimulated cells. The NETs plaque formed from stimulated cells was pre-stained with Hoechst (blue). The naïve cells (untreated cells) sat at the bottom with the presence of membrane-impermeable DNA stain SYTOX Green (green). Insets: Naïve cells closer to the NETs plaque (d2) showed greater NETosis than cells farther from the NETs plaque (d1).

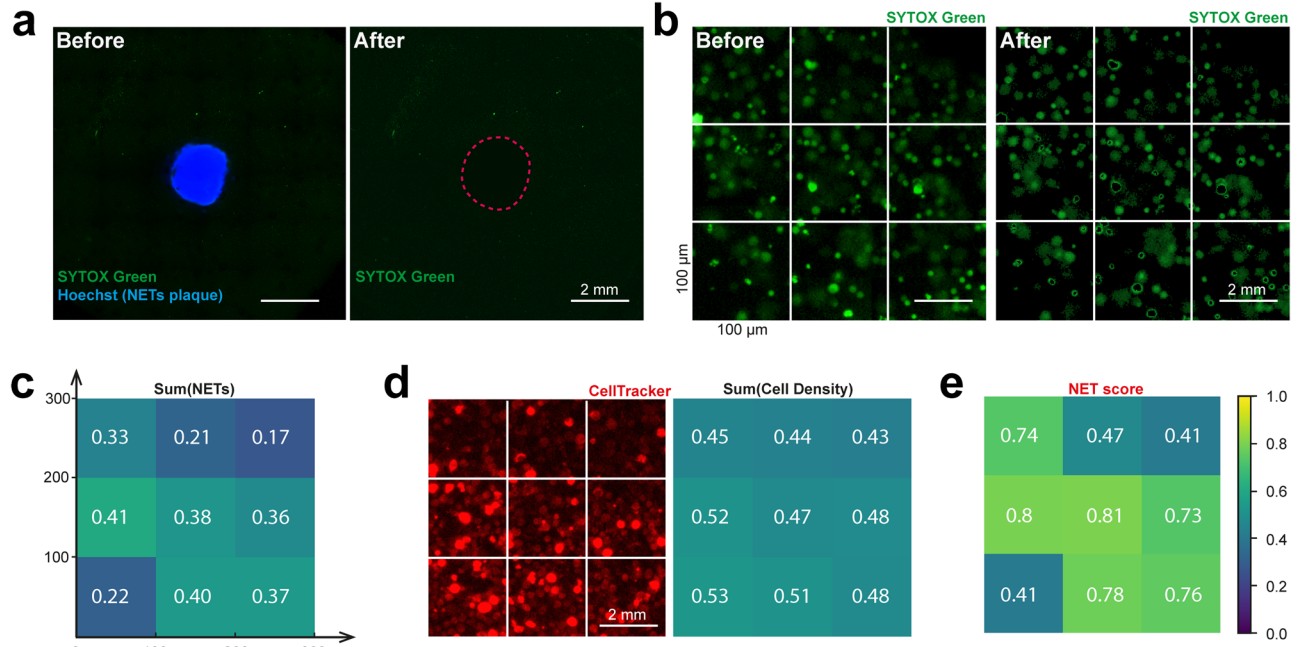

**Fig. 4 | Data analysis to obtain a spatial map of NETosis. a** Removal of pixels belonging to the NETs plaque consisting of stimulated dHL-60 cells. **b** Removal of pixels belonging to dead cells. **c** Normalized NETosis pixels in each $100 \times 100 \ \mu m^2$ square. **d** Normalized cell density in each square. **e** NET Scores of each square calculated as the ratio of NETosis and cell density measurements.

the SYTOX Green channel (Fig. 4b). These cells have permeable membranes that allow the SYTOX Green stain to penetrate, but the DNA is not spatially extended as with NETosis cells[19]. As a result, dead cells are significantly brighter and more circular than NETosis cells. To remove the dead cells, we sampled the intensity from typical dead cells and NETosis cells in each image to establish a signal threshold for removing the dead cells. After removing the NET plaque and dead cells, we process the resulting images to identify pixels containing NETs in the SYTOX Green channel, as well as cells in the CellTracker Red channel. In both cases, we measured the fluorescence intensity of regions without cells to set an intensity threshold for background pixels. The pixels for NETs and cells are then determined by all pixels that have fluorescence intensities between the threshold for the background and the threshold for dead cells. Finally, to quantify the distribution of NETosis over the entire microscopy field, we segmented the overall image into $100 \times 100 \ \mu m^2$ segments. For each segment, we calculate the ratio between the number of NETosis pixels (Green) and cell pixels (Red), which we call the NET Score (Fig. 4c–e).

To determine the rate at which NETosis is propagated from the cell plaque by secondary NETosis, we obtained the NET Score for every $100 \times 100 \ \mu m^2$ square segment over the entire microscopy field surrounding the NET plaque at different time points (Fig. 5a, Supplementary Fig. 4). We found that NET Scores increased over time in the regions close to the NET plaque (e.g. a1), but not the distant regions (e.g. a2), indicating a NET propagation at the proximal regions. This finding is consistent with our observation in fluorescent images. We observed an increasing number of NETosis pixels over time in proximal regions like region a1, and few NETosis pixels in distant regions like region a2 (Fig. 5b). To assess the propagation pattern across all regions, we averaged the NET Scores in all radial directions, at various distances from the plaque edge and at different time points (Fig. 5c). These propagation profiles showed a good fit in an inverse exponential model after 20 min ($R^2 > 0.97$):

$$NET \ score = Ae^{-Bd} + C \qquad (1)$$

where $A$, $B$, and $C$ are constants varied over time, $d$ is the distance to the edge of the NETs plaque (Fig. 5d). We also found that the increase in NET Score

diminished over time, suggesting a threshold stimulus is needed to trigger this NET propagation enabled by auto-amplification.

## Discussion

In this study, we investigated how NETosis is propagated in clusters of neutrophils in order to enhance the accumulation of NETs around the focus of infection. We used a nanowell-based assay to quantify the relationship between cell density and the proportion of neutrophils undergoing NETosis. A key advantage of this assay is that the number of cells in each nanowell could be well controlled, allowing us to establish that the percentage of cells performing NETosis is dependent on neutrophil cell density. We observed that cells treated with equal amount of ionomycin varied in NETosis frequency in a density-dependent manner, but non-induced cells exhibited a consistent baseline frequency of NETosis regardless of cell density. This phenomenon suggests that a secondary factor in cells undergoing NETosis subsequently perpetuated NETosis in those cells that were not directly induced. We examined this secondary NETosis in a nanowell-based assay by co-incubating NETosis and naïve cells. Not all treated cells secreted NETs, which may have contributed to variability in our analysis, but nonetheless, we were able to precisely correlate the number of treated cells with the frequency of NETosis in naïve cells. This observation suggests that secondary NETosis is perpetuated by cells already undergoing NETosis. We evaluated the spatial regulation of secondary NETosis using a plaque of NETs and we noted intense accumulation of NET release at the periphery of dense neutrophil clusters, which most likely enhances the containment of microbial pathogens during an infection.

The observation of secondary NETosis in neutrophil clusters was not surprising, based on recent reports that associated increased NETosis with neutrophil swarming[9–11]. For example, dense accumulation of NETs has been described in neutrophil swarming of fungal clusters[12]. Propagation of NETosis at infected sites has been attributed to interactions between neutrophils and both macrophages and platelets. However, recent evidence suggests that proteins, DNA and RNA released as part of NETs can induce NETosis in bystander naïve neutrophils[13,18,20,21]. Our results support this emerging paradigm and demonstrate that the main components of intact NETs, including DNA and NET-associated proteins, can propagate

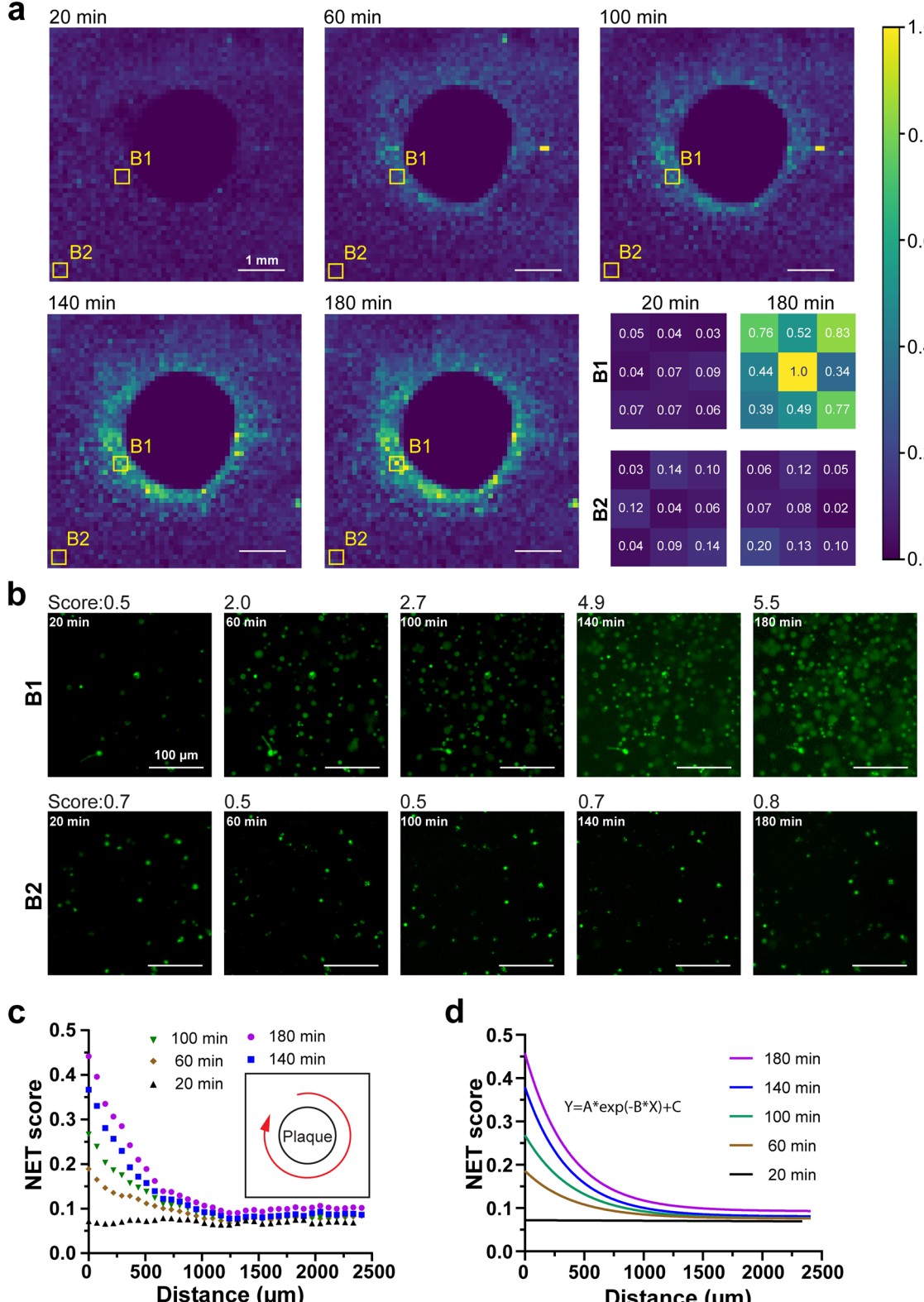

**Fig. 5 | NETosis propagation analysis. a** NET Score distributions at different time points. The NET Score was calculated from every $100 \times 100\ \mu m^2$ square segment in regions beyond a NETs plaque. Zoomed-in NET Score distributions in selected regions a1 and a2 were shown. **b** Representative SYTOX Green images of untreated cells from a proximal region (a1) and a distant region (a2). **c** NET Score averaged over all radial directions at varied radial distances from the plaque edge, at different time points. **d** NET Score in the radial directions fitted to an inverse exponential after 20 min (nonlinear regression: $R^2 > 0.97$).

NETosis among proximal naïve cells. The propagation of NETosis may be particularly relevant in circumstances where NET release by pioneer neutrophils might propagate NETosis in late-arriving cells[22]. The ability to propagate NETosis from the nidus of infection would allow secondary NETs to entrap motile pathogens that might escape from primary NETs released from pioneering neutrophils.

A surprising outcome of this study was that the observed NET propagation was spatially limited. A simple model of NET propagation resulting from a positive feedback loop powered by secondary NETosis would have caused continued propagation of NETosis to distal naïve cells so long as a high cell density is maintained. However, we instead observed propagated NETosis of naïve cells to be restricted to a thin ring surrounding the original NET plaque, despite the presence of high-density distal naïve cells. Therefore, a suppressive force must be present to dampen the secondary NETosis positive feedback loop. Potential mechanisms of such a suppressive force could include pro-resolving mediators[23,24] through a process analogous to cell signaling, which limits neutrophil swarming[25]. Potential examples of such pro-resolving mediators may include DNases or proteases released from nearby cells that could limit signal propagation by physical digestion of DNA or NET-associated proteins, respectively, and thus dampen the propagating NET signal over distance and/or time. The nanowell system described here provides a high-throughput NETosis assay that can be leveraged in future molecular screens to identify the specific factors responsible for the induction and suppression of secondary NETosis. Identification of suppressive factors may be particularly interesting because the dysregulation of NETosis auto-amplification and spatial propagation will likely exacerbate pathologies involving NETs, such as sepsis, arthritis, or cancer.

## Methods
### Cells preparation
HL-60 cells (CCL-240, ATCC) were cultured in Iscove's Modified Dulbecco's Medium (IMDM, Gibco) with 20% (v/v) heat-inactivated fetal bovine serum (FBS, Gibco) and 1% (v/v) penicillin at 37 °C in 5% $CO_2$. These cells were differentiated into neutrophil-like cells (dHL-60) by culturing with 1.3% (v/v) DMSO (Thermo Scientific) for 5 days. Human blood samples were provided by healthy donors between the ages of 18 and 60 following informed consent. The study protocol was approved by the University of British Columbia Research Ethics Board (UBC REB H20-01951). All ethical regulations relevant to human research participants were followed. To obtain purified human neutrophils and dHL-60 cells, a negative selection neutrophil isolation kit (Cat No.19666, 18000, Stemcell Technologies) was used by following the manufacturer's instructions. Purified human neutrophils and dHL-60 cells were resuspended in RPMI-1640 (Gibco) and IMDM respectively.

### NETosis induction
dHL-60 cells were pre-stained with Hoechst 33342 (Invitrogen), followed by two times of fresh medium replacement to get clean stained cells. Then, they were stimulated with 6 µM ionomycin calcium salt (Sigma-Aldrich), or 100 µg/mL LPS (Lipopolysaccharides from Escherichia coli O128:B12, Sigma-Aldrich), or 200 nM PMA (Phorbol 12-myristate 13-acetate, Sigma-Aldrich) for indicated time-periods at 37 °C in 5% $CO_2$.

### NETs visualization
After being treated with 1 µM ionomycin for 2 h, dHL-60 cells were gently washed with 1% FBS to block nonspecific protein binding. Cells were subsequently stained with Hoechst 33342 (Invitrogen), anti-Human myeloperoxidase (Invitrogen MPO455-8E6), and Alexa Fluor 647 (Santa Cruz Biotechnology SC-55549) for 30 min, followed by two washes with phosphate buffered saline (PBS, Gibco). The images were captured using an inverted fluorescence microscope (ECLIPSE Ti2-E, Nikon) with a 200× magnification and visualized using the NIS-Elements Viewer (Nikon).

### Naonowells-in-microwells fabrication
To develop our experimental nanowells-in-microwells system, we used PEGDA hydrogel due to its biocompatibility and low non-specific protein adsorption[26,27]. We first rinsed glass slides (2947-75 × 50, Corning) with acetone (Sigma-Aldrich) and then with ethanol (Sigma-Aldrich) respectively, followed by a plasma cleaning treatment to obtain a high-quality cleaning surface. The glass slides were then treated with a solution of 6% 3-(Trimethoxysilyl)propyl methacrylate (M6514, Sigma-Aldrich) at room temperature for 1 h, followed by two ethanol rinses and two distilled water rinses respectively. After that, a prepared pre-polymer solution consisting of PEGDA 250 (Sigma-Aldrich) and 1.5% (w/v) photo-initiator I819 (Sigma-Aldrich) was sprayed on these cleaned glass slides. To photopolymerize nanowells at a pitch of 80 µm × 80 µm, a UV light scanning system was used[14]. The resulting nanowells were rinsed with ethanol and distilled water to remove the uncured pre-polymers. Next, they were exposed to UV light (FH-CU-01, Formlabs) at 65 °C for 30 min for a second photopolymerization. Lastly, the glass slides patterned with nanowells were mounted to a bottomless adhesive 384-well plate (Grace Bio Labs). The final height of nanowell units ranged from 25 to 35 µm.

### Secondary NETosis assay
To prepare stimulated cells, dHL-60 cells/human primary neutrophils were stimulated by culturing with 6 µM ionomycin for at least 1 h (dHL-60 cells) and 30 min (primary neutrophils). Following incubation, the ionomycin was removed by gently resuspending stimulated cells in fresh media. They were then seeded inside nanowells at the densities indicated. The naïve cells (dHL-60/human primary neutrophils) were prepared by pre-staining with Hoechst for 20 min, followed by twice washing in fresh media. Subsequently, these naïve cells at the density of $3.5 \times 10^5$/mL (~10 nave cells/nanowell) were seeded inside nanowells in the presence of stimulated cells. The amount of secondary NETosis was quantified by averaging the percentage of NETosis in naïve cells from three randomly selected non-overlapping 5 × 5 nanowell blocks.

### Stimulated cells treatments
The stimulated primary neutrophils were prepared as described in the secondary NETosis assay but followed with another 4.5 h of incubation in ionomycin-free media. Then, these cells were either co-cultured with naïve cells as a positive control group or ready for the following treatments.

In cell-free NETs isolation, we used pipettes to generate flows to detach some NETs from stimulated cells. After centrifuging down the stimulated cells, we collected the suspension that contained cell-free NETs. In DNase I treatment on stimulated cells, we used DNase I (0.1 mg/mL, Stemcell Technologies, Canada) to digest DNA released from stimulated cells for 15 min, followed with a gentle cell washing to remove digested DNA fragments. In ODN-A151 treatment on stimulated cells, we used 5 µM ODN TTAGGG (A151) (InvivoGen) or 5 µM ODN TTAGGG Control (InvivoGen) to treat stimulated cells. In Proteinase K treatment on stimulated cells, we used Proteinase K (15 µL/mL, Cat No. P8111S, New England Biolabs) to treat the stimulated cells for 15 min at 37 °C. Then, Proteinase K was inactivated (55 °C for 10 min) followed with a temperature cooling to room temperature.

After all these treatments on stimulated cells, we co-cultured the stimulated cells with naïve cells that were pre-stained with Hoechst as described in the secondary NETosis assay. Images were taken at the beginning and the end of the coculture for the quantification of secondary NETosis.

### NETosis propagation assay
After being treated with Hoechst and 6 µM ionomycin for 4 h in a 96-well plate (Greiner Bio-One), treated dHL-60 cells in each well formed a sticky NET-rich plaque stained with Hoechst, which were further washed in fresh IMDM three times to remove residue of ionomycin. Next, the plaques were transported onto a sterilized chromatography paper (CAT No. 3030-917,

Whatman) using a pipette. Then, the papers were placed into a 24-well plate (VWR) with the side of NETs plaque facing toward naive dHL-60 cells at the well bottom. To track cell distributions over time, naive dHL-60 cells ($1.2 \times 10^6$/mL) were stained with CellTracker Deep Red Dye (1 μM, Invitrogen) before being seeded into the 24-well plate. The NETs plaque and untreated cells were incubated together in the presence of SYTOX Green Nucleic Acid Stain (1 μM, Invitrogen). They were imagined for 180 min at intervals of 40 min.

## Quantification of NETs in nanowells

Prepared human neutrophils/dHL-60 cells were first stained with Hoechst inside nanowells. Then, they were imaged immediately after the addition of indicated NETosis inducer (experimental group) or culture media (control group). After 4.5 h of incubation at 37 °C in 5% $CO_2$, cells treated with NETosis inducers underwent NETosis and showed Hoechst stains with low intensity and large area, which indicated the breakdown of cell membranes and the release of DNA beyond cell boundaries. In contrast, intact cells were rounded, small, and bright, which were used in thresholds setting in ImageJ to distinguish NETosis cells from intact cells. Since cells were confined in nanowells, the number of NETosis cells can be determined by counting the number of intact cells based on high fluorescent intensity, high circularity, and small stained area. The percent of cells undergoing NETosis (%NETs) can be calculated from the number of intact cells at the beginning ($N_1$) and end ($N_2$) of the incubation period using ($N_1 - N_2$)/ $N_1$*100%.

## Quantification of NET Score

In the NETosis propagation assay, the starting points of NETosis propagation were determined by the boundaries of the NET plaque. In other words, NETosis beyond the area of the region covered by the NET plaque were quantified. Python program was developed to quantify the NETosis propagation using NET score. Pixels positive in the Hoechst channel (blue) were first zeroed out. In the SYTOX Green channel (Green), pixels of dead cells showed higher pixel value than that of NETs, leading to a threshold value capable of distinguishing NETs from dead cells. For each image, the threshold was set based on an average of pixel values of dead cells. In the SYTOX Green channel, any pixels with values lower than the threshold and higher than the background pixels were treated as NETosis pixels. In the CellTracker Red channel, pixels positive in red were treated as cell pixels to indicate the distribution of naïve cells. Both NETosis pixels and cells pixels were normalized in $100 \times 100$ μm$^2$ segments, and the ratio between these two was the NET Score.

## Statistics and reproducibility

Experiments were performed with three independent experiments. Studies with primary neutrophils were performed with cells derived from three independent donations. NETosis frequency was compared by one-way ANOVA with Tukey's post-hoc test. Each experimental condition was repeated in at least three independent nanowell-in-microwells.

## Reporting summary

Further information on research design is available in the Nature Portfolio Reporting Summary linked to this article.

## Data availability

The data of this study are available within the paper and its Supplementary Information. The source files for all the graphs presented in the paper are 'Supplementary Data 1.xlsx' and 'Supplementary Data 2.xlsx'. All other data are available from the corresponding author (or other sources, as applicable) on reasonable request.

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

## Acknowledgements
This work was supported by grants from the Natural Sciences and Engineering Research Council of Canada (RGPIN-2020-05412, RTI-2020-00530). P.D. acknowledges funding from the China Scholarship Council and the Tai Hung Fai Charitable Foundation. A.X. acknowledges funding from MITACS (IT13817). P.M.G. is funded by a CIHR Frederick Banting and Charles Best Doctoral Award.

## Author contributions
H.M. supervised the study. H.M., K.M., P.M.G and P.D. conceived the idea. P.D. performed the experimental work. P.D. and A.X. analyzed the data. P.D, H.M., S.P.D and P.M.G wrote the manuscript.

## Competing interests
The authors declare no competing interests.
