## [Peer review file · Communications Biology]

Reviewers' comments:

Reviewer #1 (Remarks to the Author):

The study explored an interesting phenomenon where neutrophils are more susceptible to produce neutrophil extracellular traps (NETs) when they are at higher cell density or are closer to NETted cells. While the findings could add new insights to NETosis biology, the authors would need to address several key concerns to solidify their conclusion based on in vitro experiments.

(1) How representative and relevant the in vitro findings are in relation to in vivo situations. The authors experimented different cell density for NETosis but results are rather relative. For example, what is the critical cell number that NETosis is enhanced / suppressed? Is there a threshold? Figure 1A showed the seeding of cells in the nanowells – The reviewer noticed that the distances between cells were non-homogenous [i.e., in within the same well (one “cell density”, or one value for “number of cells per nanowell unit”), there were cells that were further apart while some were closer to each other]. Does number of cells per well matter or actual distance between cells matter? How does individual cell sense cell density (in 2D)? How were cell densities chosen (any reference to in vivo situation)? Figure 1D showed that as few as one cell per nanowell unit can result in 40% of NETs – What does it mean? Is the correlation in Figure 1D significant (P-value of the correlation)?

(2) Quantification of primary NETting cells. When dHL-60 cells are used, one may like to be cautious that not all cells are fully differentiated and that not all cells will undergo NETosis. The authors may consider quantifying “NETs per unit” instead of “treated cells per unit” (Figure 2) in the investigation of “paracrine signaling in NETosis”. This also circumvents the issue that some NETs may be eliminated during the two washes before addition of naïve cells.

(3) What is the paracrine factor(s) that stimulates secondary NETosis? Cytokines released from the primary NETted cells are excluded (as the preparation were washed twice before adding naïve neutrophils). The used of DNase or ODN-A151 did not abolish secondary NETosis of naïve neutrophils (only a modest reduction in NETosis is observed). The present manuscript lacks solid data to explain how auto-amplification is mediated.

(4) While the authors hypothesized that the closer the cells to the plaque, the higher is the susceptibility for the cells to undergo NETosis, the two examples (Figure 5 and Figure S4) showed that quite a large number of cells at the close periphery of the plaque were average in NET score – Why some were higher while majority were average? How is this spatially regulated (key subject matter of the manuscript)? The reviewer is also curious why B1 (Figure 5) and A1 (Figure S4) were chosen instead of other areas around the periphery. It seems that the proximal areas (B1 and A1) were chosen based on their higher NET score instead of objective sampling (i.e., increase sampling around the plaque periphery). The model built may thus result from an issue of overestimation since the NET score computed is skewed. Please revisit the analysis. What is the P-value of the current model?

(5) LPS would be a better stimulant of choice than ionomycin as the authors were trying to link the collective NETting behavior to infection. In fact, LPS induced 50% of NETosis (versus 60% when ionomycin is used) in the authors' hands (Figure S1). Suggest that some key findings to be validated using LPS.

(6) NETosis usually occurs at sites where tissue injury takes place. In addition to neutrophil DNA (e.g. NETs), other tissue DNA (released via necrosis, for example) could also be at the inflammation milieu. How would authors comment on NET auto-amplification and propagation in view of other host (or even bacterial) DNA? Is DNA the amplification trigger or more specifically NETs?

(7) Statistical analysis section (in Methods section) is missing. Are error bars showing SEM? Please also elaborate what n refers to (number of wells or average of replicas)? How many independent experiments have been performed per assay?

Reviewer #2 (Remarks to the Author):

Deng et al. investigated the interaction between ionomycin-stimulated neutrophils and nearby Naive cells. The authors found that cell density is correlated with NETosis using a nanowell system. They also displayed auto-amplification of NETosis and quantified the propagation of NETosis to proximal cells from a NET plaque. Despite the interesting data, the novelty of the work is somewhat damaged because some results are confirmatory of previously published observations including the ability of NET to induce de novo NETosis (Agarwal et al. 2019, Herster et al. 2020). Also, some of the claims would require more controls.

1. The authors used ionomycin and HL-60 to induce NETosis throughout the paper, so it is not clear if the phenotypes they have observed are model specific. They need to investigate whether other stimuli such as LPS, PMA and bacterial pathogens, and other primary neutrophil-derived NETs can lead to the same results.

2. In fig. 1, the authors demonstrated that increased cell density is associated with the frequency of NETosis by paracrine signaling. Consistently, ionomycin-induced NETosis increased in a density-dependent manner, but spontaneous NETosis was not. They need to explain what makes the difference.

3. Previous reports already demonstrated that NETs induce NETosis via TLR signaling pathways detecting DNA or RNA complex (Agarwal et al. 2019, Herster et al. 2020). The authors' suggestion that DNA release from NETs can induce secondary NETosis in nearby cells is quite confirmatory. Interestingly, in this model, DNase I and TLR inhibitor treatment just partially suppressed secondary NETosis. Which factor is mainly responsible for driving an auto-amplification of NETosis? Isolated NETs or microwebs treatment could be helpful to answer this issue.

4. Is this an ionomycin-HL60-specific event? Even though washing steps removed the majority of ionomycin to induce NETosis, isn't it possible that residual ionomycin, when they reach a certain threshold, functions as a signaling transmitter for secondary NETosis?

5. In fig. 2, the frequency of secondary NETosis increased depending on the number of ionomycin-treated cells. To investigate the role of cell density for the auto-amplification of NETosis in neutrophil swarms, the authors also need to examine if the density of naive cells affects the frequency of secondary NETosis at different time points.

6. What are the characteristics of secondary NETosis? Are they vital NETs or suicidal NETs?

7. Which factors determine the distance of NETosis propagation? Does the density of naive cells or the size and density of a NET plaque affect the distance of propagation?

Reviewer #1 (Remarks to the Author):

The study explored an interesting phenomenon where neutrophils are more susceptible to produce neutrophil extracellular traps (NETs) when they are at higher cell density or are closer to NETted cells. While the findings could add new insights to NETosis biology, the authors would need to address several key concerns to solidify their conclusion based on in vitro experiments.

We thank the reviewer for this comment and for recognizing the innovation in our work. We have substantially rewritten this manuscript and performed additional experiments to solidify our conclusions.

(1) How representative and relevant the in vitro findings are in relation to in vivo situations. The authors experimented different cell density for NETosis but results are rather relative. For example, what is the critical cell number that NETosis is enhanced / suppressed? Is there a threshold? Figure 1A showed the seeding of cells in the nanowells – The reviewer noticed that the distances between cells were non-homogenous [i.e., in within the same well (one “cell density”, or one value for “number of cells per nanowell unit”), there were cells that were further apart while some were closer to each other]. Does number of cells per well matter or actual distance between cells matter? How does individual cell sense cell density (in 2D)? How were cell densities chosen (any reference to in vivo situation)? Figure 1D showed that as few as one cell per nanowell unit can result in 40% of NETs – What does it mean? Is the correlation in Figure 1D significant (P-value of the correlation)?

We thank the reviewer for these suggestions, and we agree that the nanowell system provides an opportunity to precisely quantify NETosis at the single-cell level. Our results did not show a critical number of cells that trigger NETosis or a thresholding effect. We found that the probability of NETosis increases with cell density. The distances between cells are non-homogeneous, but the average distance between cells in each nanowell depends on the number of cells in each nanowell. For the data presented in Fig. 1D, each data point is an average of the measured % NETs from 75 nanowells (triplicate from non-overlapping 5x5 nanowells). At the minimum density, the nanowells have one or zero cell per nanowell. However, we can still obtain an averaged % NETs for these densities. We have improved our explanation of this data in our manuscript. The P-value of the correlation has been reported.

We clarified these points in our manuscript on lines 103, 112-113, 361-365.

(2) Quantification of primary NETting cells. When dHL-60 cells are used, one may like to be cautious that not all cells are fully differentiated and that not all cells will undergo NETosis. The authors may consider quantifying “NETs per unit” instead of “treated cells per unit” (Figure 2) in the investigation of “paracrine signaling in NETosis”. This also circumvents the issue that some NETs may be eliminated during the two washes before addition of naïve cells.

We agree that the differentiation of HL-60 cells to neutrophils may not be 100% efficient. Therefore, we included a neutrophil isolation step after differentiation using EasyStep Cell Separation (STEMCELL), which provides differentiated cells with >99% purity. In regards to quantifying NETosis based on “NETs per unit” rather than “treated cells per unit”, unfortunately, we do not have a simple way to quantify NETs directly. Instead, we quantified the number of treated cells per nanowell unit. The reviewer is correct in that some NETs may be removed during the washing step to remove ionomycin. However, despite this loss, we still observed secondary NETosis in the naïve cells, which shows the power of the effect.

We clarified these points in our manuscript on lines 271-273, 227-229.

(3) What is the paracrine factor(s) that stimulates secondary NETosis? Cytokines released from the primary NETted cells are excluded (as the preparation were washed twice before adding naïve neutrophils). The use of DNase or ODN-A151 did not abolish secondary NETosis of naïve neutrophils (only a modest reduction in NETosis is observed). The present manuscript lacks solid data to explain how auto-amplification is mediated.

We thank the reviewer for this important question. It is true that some cytokines released from primary NETosis cells might be removed by our washing step. We performed substantial new experiments on primary human neutrophils to investigate this question. Based on our original finding that DNase I and TLR9 inhibition partially suppressed secondary NETosis, we extracted cell-free NETs and used them to treat naïve neutrophils. Interestingly, cell-free NETs could not initiate secondary NETosis, which suggests that primary NETotic cells are required for secondary NETosis. We then used Proteinase K to digest protein components associated with stimulated cells, which also eliminated secondary NETosis. This result indicates that secondary NETosis requires a combination of DNA and protein factors from proximal NETotic cells.

We clarified these points in our manuscript on lines 136-155. Please see the new data figure below.

Fig. 2D Secondary NETosis in primary neutrophils after different treatments on stimulated cells: stimulated cells (no treatment), cell-free NETs isolation, ODA-A151 treatment, DNase I treatment, Proteinase K treatment.

(4) While the authors hypothesized that the closer the cells to the plaque, the higher is the susceptibility for the cells to undergo NETosis, the two examples (Figure 5 and Figure S4) showed that quite a large number of cells at the close periphery of the plaque were average in NET score – Why some were higher while majority were average? How is this spatially regulated (key subject matter of the manuscript)? The reviewer is also curious why B1 (Figure 5) and A1 (Figure S4) were chosen instead of other areas around the periphery. It seems that the proximal areas (B1 and A1) were chosen based on their higher NET score instead of objective sampling (i.e., increase sampling around the plaque periphery). The model built may thus result from an issue of overestimation since the NET score computed is skewed. Please revisit the analysis. What is the P-value of the current model?

I think our previous manuscript did not fully explain our work here. We analyzed all regions surrounding a NET plaque to obtain a NET score in each region. We then plotted these NET scores to show the distribution of NET scores around the NET plaque. We found the NET plaque was surrounded by a ring of regions with higher NET score and that the NET score decreased radially in an exponential manner. The regions B1(or A1) and B2 (or A2) were selected to show examples of regions with high and low NET scores.

We've clarified our explanations in the revised manuscript on lines 200-214.

(5) LPS would be a better stimulant of choice than ionomycin as the authors were trying to link the collective NETting behavior to infection. In fact, LPS induced 50% of NETosis (versus 60% when ionomycin is used) in the authors' hands (Figure S1). Suggest that some key findings to be validated using LPS.

In response to the reviewer's suggestion, we performed additional experiments using LPS to investigate the relationship between the cell density and %NETs (Fig. 1E). We treated human (primary) neutrophils with 20 µg/mL of LPS and measured the percentage of cells undergoing NETosis (%NETs) under different cell density. The result exhibited a positive correlation between cell density and the %NETs (P value <0.001, $R^2=0.767$), which is consistent with the results obtained from dHL-60 cells treated with ionomycin (Fig. 1D). Therefore, we conclude that increased cell density is associated with increased potential for NETosis, which suggests the potential for paracrine signalling between cells to propagate NETosis.

We have clarified these points in our manuscript on lines 112-113.

Fig. 1. NETosis depends on cell density. **(D)** Percentage of dHL-60 cells producing NETs after ionomycin stimulation (6 µM, 4.5 h) relative to the averaged density of total cells incubated in nanowells. **(E)** Percentage of human neutrophils producing NETs after ionomycin (5 µM, 4.5 h) or LPS stimulation (20 µg/mL, 4.5 h) relative to the averaged density of total cells incubated in nanowells. All P values < 0.001.

(6) NETosis usually occurs at sites where tissue injury takes place. In addition to neutrophil DNA (e.g. NETs), other tissue DNA (released via necrosis, for example) could also be at the inflammation

milieu. How would authors comment on NET auto-amplification and propagation in view of other host (or even bacterial) DNA? Is DNA the amplification trigger or more specifically NETs?

We thank the reviewer for this comment. In this study, we are primarily interested in the secondary NETosis induced by neutrophils undergoing NETosis. Existing findings show that mitochondrial DNA (mtDNA) from trauma tissues could induce NETosis through TLR9 signalling pathways.¹ Bacterial DNA could also contribute to NETosis due to the similarity between them with mtDNA.

(7) Statistical analysis section (in Methods section) is missing. Are error bars showing SEM? Please also elaborate what n refers to (number of wells or average of replicas)? How many independent experiments have been performed per assay?

We thank the reviewer for raising this issue. For secondary NETosis assays, the values shown are a mean of three individual experiments, involving different donors. Error bars represent standard deviation. N refers to the number of individual microwells. The %NETs in each microwell were averaged from three randomly selected non-overlapping 5×5 nanowell blocks. We have added a statistical methods section where this is described and have modified the figures to clarify that standard deviation was used.

We have clarified these points in our manuscript on lines 309-311, 361-366.

References:

1. Liu, Li, et al. "Induction of neutrophil extracellular traps during tissue injury: Involvement of STING and Toll-like receptor 9 pathways." *Cell proliferation* 53.10 (2020).

Reviewer #2 (Remarks to the Author):

Deng et al. investigated the interaction between ionomycin-stimulated neutrophils and nearby Naive cells. The authors found that cell density is correlated with NETosis using a nanowell system. They also displayed auto-amplification of NETosis and quantified the propagation of NETosis to proximal cells from a NET plaque. Despite the interesting data, the novelty of the work is somewhat damaged because some results are confirmatory of previously published observations including the ability of NET to induce de novo NETosis (Agarwal et al. 2019, Herster et al. 2020). Also, some of the claims would require more controls.

We thank the reviewer for recognizing the contributions of our study. Our work is novel in the following three ways: 1) We developed a high-throughput assay for studying NETosis and used it to show that NETosis depends on cell density. 2) We confirmed the existence of secondary NETosis and we showed the mechanism depends on insoluble DNA and proteins in the released NETs. 3) We developed a method to analyze the spatial propagation of secondary NETosis and showed that this process is self-regulated.

1. The authors used ionomycin and HL-60 to induce NETosis throughout the paper, so it is not clear if the phenotypes they have observed are model specific. They need to investigate whether other stimuli such as LPS, PMA and bacterial pathogens, and other primary neutrophil-derived NETs can lead to the same results.

To prove that our observations of secondary NETosis do not exist only in ionomycin treatment on dHL-60 cells, we performed additional experiments on human neutrophils to investigate the relationship between the cell density and %NETs (**Fig. 1E**). The human (primary) neutrophils at different cell densities were treated either with 20 $\mu\text{g}/\text{mL}$ of LPS or 5 μM of ionomycin. As expected, their NETosis quantification results showed a positive correlation between cell density and the %NETs (P value <0.001 , $R^2=0.767$ in ionomycin treatment, $R^2=0.691$ in LPS treatment), which is consistent with the results obtained from dHL-60 cells treated with ionomycin (**Fig. 1D**). Therefore, we conclude that increased cell density is associated with increased potential for NETosis, which suggests the potential for paracrine signalling between cells to propagate NETosis.

We also revised our manuscript on lines 112-113. Please see the revised Figure 1 below.

Fig. 1. NETosis depends on cell density. (D) Percentage of dHL-60 cells producing NETs after ionomycin stimulation (6 μ M, 4.5 h) relative to the averaged density of total cells incubated in nanowells. (E) Percentage of human neutrophils producing NETs after ionomycin (5 μ M, 4.5 h) or LPS stimulation (20 μ g/mL, 4.5 h) relative to the averaged density of total cells incubated in nanowells. All P values < 0.001.

2. In fig. 1, the authors demonstrated that increased cell density is associated with the frequency of NETosis by paracrine signaling. Consistently, ionomycin-induced NETosis increased in a density-dependent manner, but spontaneous NETosis was not. They need to explain what makes the difference.

We thank the reviewer for this comment. Ionomycin-induced (and LPS-induced) NETosis increased in a density-independent manner because some of the NETosis cells induced secondary NETosis, which increased the total number of NETosis cells. In the control sample, there were very few NETosis cells, which means there are almost no induced secondary NETosis to increase the total number of NETosis cells.

3. Previous reports already demonstrated that NETs induce NETosis via TLR signaling pathways detecting DNA or RNA complex (Agarwal et al. 2019, Herster et al. 2020). The authors' suggestion that DNA release from NETs can induce secondary NETosis in nearby cells is quite confirmatory. Interestingly, in this model, DNase I and TLR inhibitor treatment just partially suppressed secondary NETosis. Which factor is mainly responsible for driving an auto-amplification of NETosis? Isolated NETs or microwebs treatment could be helpful to answer this issue.

We thank the reviewer for this comment. We performed substantial new experiments on primary human neutrophils to investigate this question. Based on our original finding that DNase I and TLR9 inhibition partially suppressed secondary NETosis, we extracted cell-free NETs and used them to treat naïve neutrophils. Interestingly, cell-free NETs could not initiate secondary NETosis, which suggest that primary NETotic cells are required for secondary NETosis. We then used Proteinase K to digest protein components associated with stimulated cells, which also eliminated secondary NETosis. This result indicates that secondary NETosis requires a combination of DNA and protein factors from proximal NETotic cells.

We clarified these points in our manuscript on lines 136-155. Please see the new figure below.

Fig. 2D Secondary NETosis in primary neutrophils after different treatments on stimulated cells: stimulated cells (no treatment), cell-free NETs isolation, ODA-A151 treatment, DNase I treatment, Proteinase K treatment.

4. Is this an ionomycin-HL60-specific event? Even though washing steps removed the majority of ionomycin to induce NETosis, isn't it possible that residual ionomycin, when they reach a certain threshold, functions as a signaling transmitter for secondary NETosis?

To address this concern, we performed additional experiments using primary neutrophils, which were induced to NETosis using ionomycin and LPS (**Fig. 1E**). We also used primary neutrophils to investigate the auto-amplified NETosis (**Fig. 2C**). All results obtained from human neutrophils corresponded with dHL-60 cell experiments, suggesting that our observations are not dHL-60 cell-specific. With regards to

the question about residual ionomycin, we treated naïve cells with the wash supernatant. We found that the wash supernatant did not contain enough ionomycin to trigger NETosis.

We clarified these points in our manuscript on lines 112-113, 125-126, 130-133.

5. In fig. 2, the frequency of secondary NETosis increased depending on the number of ionomycin-treated cells. To investigate the role of cell density for the auto-amplification of NETosis in neutrophil swarms, the authors also need to examine if the density of naive cells affects the frequency of secondary NETosis at different time points.

We thank the reviewer for this comment. We used naïve cells as a consistent reporter for factors released by stimulated cells that result in secondary NETosis. While it is true that a greater density of naïve cells would further increase the amount of secondary NETosis, this effect could potentially confound our experiments. Therefore, we used a constant number of naïve cells in all our experiments.

6. What are the characteristics of secondary NETosis? Are they vital NETs or suicidal NETs?

We thank the reviewer for raising this question. In the NETosis propagation assay, we used SYTOX Green to visualize secondary NETosis propagated from the NETs plaque. SYTOX Green is a dead cell indicator that is impermeable to live cell membranes. After SYTOX Green staining, we observed NETosis cells positive in SYTOX Green, suggesting the presence of suicidal NETosis in secondary NETosis.

7. Which factors determine the distance of NETosis propagation? Does the density of naive cells or the size and density of a NET plaque affect the distance of propagation?

Our nanowell-based assay revealed that both protein and DNA are important factors in NETosis propagation. However, identifying factors that affect propagation in plaques is more challenging. The plaque assay monitors cell distance in 3D, which makes it prohibitively difficult to control plaque size or determine precise cell density. In future work, we aim to develop new methods for more precise plaque production.

REVIEWERS' COMMENTS:

Reviewer #1 (Remarks to the Author):

Reviewer's concerns are adequately addressed.

Reviewer #2 (Remarks to the Author):

The manuscript by Deng and co-workers is improved by revision. I do have a few points that I believe should be addressed, but I will leave it to the discretion of the handling editor.

Fig. 3 indicates that the numbers of SYTOX+ cells do not show significant differences between Proximal vs Distal or plaque-treated vs control. However, there is a distance-dependent relationship observed in NET score and propagation. This suggests that NET formation and cell death might not be tightly linked. Also, it is challenging to differentiate dead cells from NET+ cells in the figures provided. To gain better insights, it's necessary for the authors to present cell tracker staining images that they have with these images and merged images. Moreover, as vital NETs can be stained by SYTOX because of extruded DNA, the authors' response to the reviewer's previous question seems insufficient.

Reviewer #1 (Remarks to the Author):

Reviewer's concerns are adequately addressed.

We thank the reviewer for reviewing our revised manuscript.

Reviewer #2 (Remarks to the Author):

The manuscript by Deng and co-workers is improved by revision. I do have a few points that I believe should be addressed, but I will leave it to the discretion of the handling editor.

We thank the reviewer for recognizing improvement in our manuscript. Please see our response below.

Fig. 3 indicates that the numbers of SYTOX+ cells do not show significant differences between Proximal vs Distal or plaque-treated vs control. However, there is a distance-dependent relationship observed in NET score and propagation. This suggests that NET formation and cell death might not be tightly linked. Also, it is challenging to differentiate dead cells from NET+ cells in the figures provided. To gain better insights, it's necessary for the authors to present cell tracker staining images that they have with these images and merged images. Moreover, as vital NETs can be stained by SYTOX because of extruded DNA, the authors' response to the reviewer's previous question seems insufficient.

The reviewer is right that the number of SYTOX+ cells is not tightly linked to the number of NETosis cells in our assay. This is because SYTOX+ cells are dead cells that retain their DNA, whereas NETs are characterized by a weaker diffuse fluorescent signal. We characterize the spatial distribution of NETs by first excluding pixels associated with the SYTOX+ dead cells. We then determine the NET Score over a given area from the ratio of NET pixels to the number of cell pixels. Using this analysis, we can look at the NETosis propagation from a NET plaque (Fig. 3d), and we can see that the amount of NETs in the proximal region d2 is significantly greater than in the distal region d1. Finally, regarding the reviewer's comment about vital NETs, both vital and non-vital NETs could be stained by SYTOX and were captured in our analysis.

We made some minor adjustments to Figure 3 and 5, as well as to the results section to help clarify some of these issues.